# Prevalence and Determinants of Malnutrition in Community-Dwelling Adults Aged 65 and over in Eastern Türkiye: A Cross-Sectional Study

**DOI:** 10.3390/nu17152522

**Published:** 2025-07-31

**Authors:** Emine Kemaloğlu, Betül Çiçek, Melih Kaan Sözmen, Mehmetcan Kemaloğlu

**Affiliations:** 1Department of Nutrition and Dietetics, Faculty of Health Sciences, Ağrı Ibrahim Cecen University, 04100 Ağrı, Türkiye; mkemaloglu@agri.edu.tr; 2Department of Nutrition Dietetics, Institute of Health Sciences, Erciyes University, 38260 Kayseri, Türkiye; 3Department of Nutrition and Dietetics, Faculty of Health Sciences, Erciyes University, 38260 Kayseri, Türkiye; 4Department of Public Health, Faculty of Medicine, Izmir Katip Celebi University, 35620 Izmir, Türkiye; melihkaan.sozmen@ikcu.edu.tr

**Keywords:** anthropometry, cognitive function, malnutrition, older adults, physical performance

## Abstract

**Background/Objectives:** Malnutrition in older adults is both preventable and treatable, yet its detection and etiology remain complex. Therefore, the aim of this study was to evaluate the prevalence of malnutrition and various factors involved in the etiology of malnutrition in community-dwelling individuals aged 65 years and older. **Methods:** This cross-sectional study was conducted with community-dwelling individuals aged 65 years and older in a health center in Ağrı, Türkiye. The nutritional status of older adults was measured using the Mini Nutritional Assessment (MNA). Data were collected through face-to-face interviews and a series of validated instruments, including the Standardized Mini Mental Examination (MMSE), body composition measurements (BIA), dietary intake records, and physical performance tests such as hand grip strength, chair stand, and Timed ‘Up & Go’ (TUG) Test. Statistical analyses included chi-square and Mann-Whitney U tests for group comparisons and logistic regression to investigate independent factors associated with risk of malnutrition. **Results:** A total of 182 participants were included in the study. The mean age of the participants was 72.1 ± 6.0 years. Of the participants, 59.3% were male. 1.6% of the participants were malnourished, and 25.3% were at risk of malnutrition. Perceived health status compared to peers (OR: 1.734, 95% CI: 1.256–2.392, *p* = 0.001), lower appetite status (OR: 1.942, 95% CI: 1.459–2.585, *p* < 0.001) and lower waist circumference (OR: 1.089, 95% CI: 1.040–1.140, *p* < 0.001) were independent predictors of malnutrition risk. **Conclusions:** The risk of malnutrition was higher among individuals with lower appetite, poorer self-perceived health status compared to peers, and smaller waist circumference. Reduced physical function and strength were also associated with an increased risk of malnutrition.

## 1. Introduction

According to the World Health Organization (WHO), people are considered to be elderly from the age of 65 in developed countries, and from 60 in developing countries [1]. According to WHO data, there were 1 billion people aged 60 and over in the world in 2019, and this number is expected to increase to 1.4 billion in 2030 and 2.1 billion in 2050 [2]. The elderly population is also increasing in Türkiye, and by 2024, the population aged 65 and over will constitute 10.6% of the total population [3].

The European Society for Clinical Nutrition and Metabolism (ESPEN) defines malnutrition as ‘a state resulting from lack of intake or uptake of nutrition that leads to altered body composition (decreased fat-free mass) and body cell mass leading to diminished physical and mental function and impaired clinical outcome from disease’ [4]. Older adults are prone to age-related diseases, physical changes, and functional impairments that may lead to malnutrition [5]. Malnutrition is a preventable and treatable condition if detected early. However, many elderly individuals living in the community still experience malnutrition and its associated physical and mental impairments [6].

Factors involved in the etiology of malnutrition differ for elderly individuals staying in hospitals, nursing homes, and living in the community [7,8,9]. Many factors, such as individual, environmental, and disease-related factors, play a role in the etiology of malnutrition in individuals living in the community [10]. Although there are non-modifiable risk factors for malnutrition, such as age and gender, many factors involved in the etiology of malnutrition are modifiable [7]. Income, living space, malnutrition, and social security are important risk factors that can be changed by policies for the elderly [11,12]. Several factors, including education level, anthropometric characteristics, health status, physical function, multimorbidity, polypharmacy, and cognitive impairment, are key risk factors requiring lifestyle changes, and ideally, should have been addressed in earlier stages of life [13,14]. Detection of malnutrition in elderly individuals living in the community is crucial in terms of reducing morbidity and mortality rates and reducing the individual and social burden due to malnutrition. In addition, identifying risk factors that may lead to malnutrition in later life is crucial for informing long-term health policies and promoting healthy lifestyle changes from a younger age to reduce the risk of malnutrition in later life. While several studies have investigated malnutrition in institutionalized or hospitalized elderly individuals, data focusing on community-dwelling older adults in Türkiye remain limited. This study aims to fill this gap by examining both the prevalence and modifiable risk factors of malnutrition in this population, thereby contributing valuable insight for public health planning and preventive strategies. 

## 2. Materials and Methods

### 2.1. Study Design and Participants

This cross-sectional study was conducted in a family health center in Ağrı, located in the eastern province of Türkiye, with individuals aged 65 years and older. Data were collected from individuals who visited the family health center between August 2024 and March 2025, met the inclusion criteria, and volunteered. Information on inclusion and exclusion criteria is given in Table 1.

As the data were collected within the scope of a simultaneous scale development study, the sample size was determined based on standard recommendations for factor analysis, with a minimum participant-to-item ratio of 10:1. Given that the scale included 12 items, a minimum of 120 participants was targeted, and 182 were ultimately included. Therefore, no separate a priori sample size calculation was performed for this secondary analysis. To confirm adequacy, a post-hoc power analysis was conducted. Post-hoc power analysis was performed on the energy intake value, which is one of the important determinants of malnutrition. Considering 1465.0 ± 486.8 kcal in the group at risk of malnutrition (*n* = 45) and 1758.8 ± 574.0 kcal in the group with normal nutritional status (*n* = 132), a two-way post hoc power analysis revealed a power of 88.8%. This value shows that the sample has sufficient power to determine the differences associated with malnutrition.

The dependent variable in this study was malnutrition risk. Independent variables included demographic characteristics, cognitive performance, anthropometric measurements, dietary intake assessments, and physical function tests.

### 2.2. Demographic and Health-Related Characteristics

Information about the demographic data of the participants was obtained with a questionnaire form prepared by the researchers after reviewing the relevant literature. In addition to age and sex, the following variables were included:

Education level was assessed by asking whether participants were illiterate, literate without formal education, or had completed primary, secondary, or college education.

Marital status was categorized as married, divorced, separated, widowed, or never married.

Living arrangements were evaluated by asking with whom the participant had been living during the past six months (e.g., alone, with spouse, with children, or extended family).

Income level was self-reported and categorized based on perceived financial status as: income > expenses, income = expenses, or income < expenses.

Social security status was determined by asking participants about their current health coverage. Responses were grouped as: Green card (state-funded health insurance for low-income individuals in Türkiye), or other (including all contributory or privately paid insurance systems).

Perceived health status, appetite, and pain were recorded using the Visual Analog Scale (VAS), numbered from 1 to 10. Perceived health status was rated as 1: very poor to 10: very good, pain was rated as 1: no pain to 10: unbearable pain, and appetite was rated as 1: very poor to 10: very good.

### 2.3. Malnutrition Assessment

Malnutrition status was assessed with the Turkish version of the Mini Nutritional Assessment (MNA^®^, Ankara, Turkey) tool. The total score that can be obtained from the MNA ranges from 0–30; <17 points are considered as ‘malnutrition’, 17–23.5 points as ‘malnutrition risk’, and 24–30 points as ‘well-nourished’ [15].

### 2.4. Cognitive Performance

The Standardized Mini-Mental Test (MMSE) was used to assess cognitive performance. The range of scores that can be obtained from the scale, consisting of a total of 11 items, is 0–30; scores between 0–12, 13–22, and 23–24 indicate severe, moderate, and mild cognitive impairment. In the range of 25–30 points, there is no cognitive impairment [16,17].

### 2.5. Anthropometric Measures

Body weight, body fat, and muscle percentage were measured using a Tanita SC 330 (Tokyo, Japan) branded bioelectrical impedance (BIA) device. Height was measured using a stadiometer. Mid-Upper Arm Circumference (MUAC), waist, and calf circumference were measured with a non-flexible tape measure. Body Mass Index (BMI) was calculated by dividing body weight by the square of height.

### 2.6. Dietary Intake Assessment

Dietary intake was retrospectively assessed using a 24-h recall method, in which participants were asked to recall all foods and beverages consumed over the previous day, including estimated portion sizes. The collected data were analyzed using the Computer-Aided Nutrition Analysis Program [Nutrition Information System (BeBiS)]. Reported energy intakes of less than 800 kcal/day for men and less than 500 kcal/day for women, as well as intakes exceeding 4000 kcal/day for men and 3500 kcal/day for women, were excluded from the analysis in order to reduce the impact of outliers on statistical analysis to the results [18].

### 2.7. Physical Function Tests

The 30-s Chair-Stand Test (30-CST) is a test that evaluates the sit-to-stand activity, lower limb strength, and dynamic balance of the individual. The number of times the patient sits up within 30 s gives the score of the test. Less than 10 sit-ups in 30 s indicates lower limb weakness [19].

The Timed ‘Up & Go’ Test (TUG Test) assesses fall risk and mobility in older people. The test is performed by the individual getting up from the chair, walking three meters, coming back, and sitting back on the chair. If the elderly person completes this test in more than 12 s, they are considered at risk of falls [20].

In addition, hand grip strength was measured with a Camry brand electronic hand dynamometer (Shenzhen, China) to evaluate physical function. The cut-off points used for EWGSOP 2 sarcopenia were taken as reference for hand grip strength. Accordingly, <16 for women and <27 for men were associated with weakness in physical function [21].

### 2.8. Statistical Analysis

The data obtained were analyzed with the Statistical Package for the Social Sciences SPSS (Version 25.0. Armonk, NY, USA). Missing and/or outlier data were cross-checked with the original questionnaire and corrected before analysis. Since the number of participants with malnutrition was low, these participants were combined with the group at risk of malnutrition. The normal distribution of quantitative variables was evaluated by the Kolmogorov-Smirnov test. Descriptive statistics of quantitative variables were given as mean ± standard deviation (x ± SD), and descriptive statistics of categorical variables were given as frequency and percentage. In intergroup comparisons, the Mann-Whitney U test was used for continuous variables where normality was not provided, and the Pearson Chi-square test was used for categorical variables.

In order to determine the factors associated with the risk of malnutrition, univariate logistic regression analyses were first performed, and the variables that were significant at the *p* < 0.10 level were included in the multivariate model. In multiple logistic regression analysis, age and gender were included in the model as the main variables in the first step, and then other variables were added by the stepwise block method. Since energy and protein intake were also included, multiple regression analysis was performed with 177 participants. Thus, the relationship of each variable with the risk of malnutrition was evaluated regardless of the effect of age and gender. *p* < 0.05 values were considered statistically significant.

### 2.9. Institutional Review Board Statement

Ethical approval for this study was obtained from the Scientific Research Ethics Committee of Ağrı İbrahim Çeçen University (approval number: 276, dated 30 November 2023). Additionally, institutional permission to conduct the study at the family health center was granted by the Ağrı Provincial Health Directorate.

## 3. Results

### 3.1. Characteristics of the Study Participants

A total of 182 participants were included in the analysis, and the participation rate was 86.3%. The flowchart for the inclusion of participants in the study is given in Figure 1. The mean age was 72.1 ± 6.0 years. Of the individuals included, 108 (59.3%) were male and 148 (81.3%) were married. Variables related to the demographic data of the participants are given in Table 2.

### 3.2. Prevalence of Malnutrition Among Elderly People

According to the MNA, 3 (1.6%) of the individuals were malnourished, 46 (25.3%) were at risk of malnutrition, and 133 (73.1%) had normal nutritional status. Due to the small number of malnourished individuals (*n* = 3), this group was combined with those at risk of malnutrition to increase statistical power and ensure reliable analysis. This combined group was referred to as the “malnutrition risk” group (MNA score < 23.5).

### 3.3. Differences in Sociodemographic Characteristics by Nutritional Status

Differences in sociodemographic characteristics by nutritional status are presented in Table 2. The mean age of individuals at risk of malnutrition was higher (73.8 ± 6.6) compared to those with normal nutrition (71.5 ± 5.6) (*p* = 0.045). Female gender was significantly more frequent in the malnutrition risk group (29/49, 59.2%) than in the normal nutrition group (45/133, 33.8%) (*p* = 0.004). Similarly, low educational level (illiterate: 21/49, 42.9% vs. 25/133, 18.8%, *p* = 0.003) and having green card insurance (17/49, 34.7% vs. 21/133, 15.8%, *p* = 0.010) were more common in the malnutrition risk group.

### 3.4. Differences in Clinical, Nutritional, and Functional Indicators by Nutritional Status

Differences in clinical, nutritional, and functional indicators according to nutritional status are presented in Table 3. Participants at risk of malnutrition reported poorer perceived health (VAS: 5.1 ± 1.8 vs. 6.5 ± 1.9, *p* < 0.001), lower appetite (VAS: 4.9 ± 2.2 vs. 7.6 ± 1.6, *p* < 0.001), and more frequent pain (VAS: 5.9 ± 2.6 vs. 4.3 ± 2.9, *p* = 0.001). In terms of anthropometric measures, individuals at risk had significantly lower values for BMI (29.1 ± 6.3 kg/m^2^ vs. 31.4 ± 5.6 kg/m^2^, *p* = 0.034), waist circumference (96.6 ± 12.6 cm vs. 105.9 ± 11.6 cm, *p* < 0.001), muscle mass (43.5 ± 6.4 kg vs. 51.0 ± 8.5 kg, *p* < 0.001), and MUAC (27.8 ± 3.9 cm vs. 29.4 ± 3.6 cm, *p* = 0.009).

Regarding dietary intake, both energy (1465.0 ± 486.8 kcal vs. 1758.8 ± 574.0 kcal, *p* = 0.003) and protein (65.8 ± 24.2 g vs. 79.5 ± 37.1 g, *p* = 0.024) intakes were significantly lower among those at risk of malnutrition. Cognitive impairment was also more prevalent in the at-risk group, with 28.6% experiencing significant impairment compared to 8.3% in the well-nourished group (*p* < 0.001). In terms of physical function, handgrip strength was poor among those at risk (51.0% vs. 29.3%, *p* = 0.011), and 83.7% of this group showed poor performance in the 30-s chair stand test compared to 66.7% in the well-nourished group (*p* = 0.039).

### 3.5. Independent Predictors of Malnutrition Risk: Multivariate Logistic Regression Analysis

Multivariate logistic regression analysis of independent determinants of malnutrition risk, adjusted for age and gender, is presented in Table 4. In univariate analysis, several factors, including marital status, perceived health, perceived appetite, BMI, muscle mass, waist circumference, MUAC, and energy and protein intake, were significantly associated with malnutrition risk (*p* < 0.05). However, in the age- and sex-adjusted multivariate logistic regression model, only a few of these factors remained statistically significant, including perceived health status compared to peers, lower appetite status, and smaller waist circumference. Perceived health status compared to peers (OR: 1.73, 95% CI: 1.26–2.39, *p* = 0.001), lower appetite status (OR: 1.94, 95% CI: 1.46–2.59, *p* < 0.001), and smaller waist circumference (OR: 1.09, 95% CI: 1.04–1.14, *p* < 0.001) emerged as significant independent predictors of malnutrition risk. Handgrip strength was an independent predictor of malnutrition risk with marginal statistical significance (OR: 1.07, 95% CI: 1.00–1.15, *p* = 0.057).

## 4. Discussion

The aim of this study was to investigate the prevalence of malnutrition risk and determinants of malnutrition in community-dwelling older adults. Anthropometric data, nutritional status, physical and cognitive function, perceived health, and environmental parameters that may constitute risk factors for malnutrition risk were evaluated. According to our results, 26.9% of the participants were at risk of malnutrition. Although our results are consistent with many studies in the literature [22,23,24] reported prevalence of malnutrition varies due to several factors such as the study setting, applied exclusion criteria, and the mean age of the participants [25,26,27,28].

Our participants’ age distribution aligns with previous studies [29,30,31]. However, unlike many studies reporting a higher proportion of female participants [29,32,33,34,35], our sample had more males. This discrepancy might reflect regional cultural and linguistic factors affecting older women’s willingness or ability to participate, which could have implications for the generalizability of our findings to female populations in similar settings. When our results were analyzed, the most important independent determinant of malnutrition risk was appetite status, as measured with the VAS scale. Low appetite increased the risk of malnutrition by 1.9 times. Our findings are consistent with other studies in the literature [1,9,22]. The etiology of poor appetite in older adults is complex and may be influenced by a range of factors, including physiological changes, diseases, and drug use [36,37]. Loss of appetite is a significant risk factor for malnutrition, as it is directly associated with reduced food intake [38]. The strong effect observed here emphasizes the importance of routine assessment of appetite in older adults as part of nutritional screening. This screening method, which can be performed quickly and without additional cost, can play an important role in predicting and preventing malnutrition risk.

Another important predictor identified in this study was perceived health status compared to peers, assessed using the VAS scale. According to our results, perceiving one’s health as worse than that of peers increased the risk of malnutrition by 1.7 times. This result is consistent with previous findings in the literature [27,39]. A study reported that lower perceived health status is associated with greater feelings of loneliness and depressive mood among older adults [40]. This suggests that poor perceived health status may be indirectly linked to malnutrition as well. These findings highlight the significance of self-rated health in identifying malnutrition risk. Our study underscores that simple, subjective assessment tools may serve as early warning indicators of geriatric malnutrition.

Among the anthropometric variables assessed, waist circumference emerged as a significant predictor of malnutrition risk in the multivariate logistic regression model. Although initial analyses suggested a protective effect of higher waist circumference, this relationship reversed after adjustment for other variables. This finding indicates that waist circumference is entangled in complex interactions with other factors and should not be interpreted in isolation.

While waist circumference is commonly associated with abdominal obesity and metabolic disorders [41] in older adults, it may also reflect reduced lean muscle mass and has been linked to increased mortality risk [42]. Moreover, when considering the obesity paradox in this population [43], both BMI and waist circumference should be interpreted with caution in the geriatric context. Therefore, waist circumference should be evaluated alongside other variables and interpreted carefully when assessing nutritional risk in older adults.

According to our findings, physical performance tests such as the 30s-CST and the TUG Test, which assess functional capacity, were independently associated with malnutrition risk. Completing fewer than 10 repetitions in the 30s-CST and taking longer than 12 s in the TUG test increased the risk of malnutrition by 1.4 and 1.3 times, respectively.

In addition, handgrip strength, a commonly used clinical indicator of muscle strength, was almost significant predictor of malnutrition risk. Consistent with our results, numerous studies in the literature have shown that indicators of physical performance and muscle strength are associated with malnutrition risk in older adults [44,45,46]. Although physical function tests are primarily used in sarcopenia screening, they may also be considered practical and cost-effective tools for malnutrition screening in older adults.

Finally, variables such as muscle mass, MUAC, protein intake, and cognitive performance were significantly associated with malnutrition risk in the univariate logistic regression analysis; however lost their significance in the multivariate model. This may be due to interrelationships among these variables. For example, muscle mass, MUAC, and protein intake may indirectly affect malnutrition risk by influencing physical function and strength. Alternatively, the presence of stronger predictors in the model may have masked the contribution of weaker variables. Nonetheless, this does not diminish the potential clinical importance of variables that showed statistical significance in univariate analysis. These findings suggest that such factors should still be carefully evaluated when assessing malnutrition risk in older adults.

The findings of this study have several practical implications for clinicians, public health professionals, and policymakers. Given the strong predictive value of appetite status and perceived health, simple and low-cost screening tools such as VAS can be easily incorporated into routine geriatric assessments in community settings. In addition, the associations between physical performance measures and malnutrition risk underscore the importance of integrating functional assessments like the 30s-CST and TUG test into nutritional evaluations. Public health strategies aimed at preventing malnutrition in older adults should focus on early detection using accessible tools, promotion of physical function, and support for subjective well-being. Furthermore, targeting modifiable risk factors through individualized interventions and community-based programs could help reduce the burden of malnutrition and improve quality of life in aging populations. Since our study is cross-sectional in nature, it limits the ability to infer causality. Additionally, reliance on self-reported data constitutes another important limitation. However, to minimize the potential impact of varying levels of comprehension among participants, all survey questions were administered directly by the researchers through one-on-one interviews. This approach allowed the inclusion of illiterate participants and helped reduce the risk of misinterpretation or inaccurate responses. It also contributed to minimizing missing data in the dataset.

We believe that studies including not only community-dwelling older adults, but also those residing in hospitals or nursing homes, face challenges in terms of group homogeneity. To better observe the influence of environmental factors on malnutrition, our study specifically focused on community-dwelling older adults who did not have serious underlying conditions that could directly cause malnutrition. We believe this approach enhanced the homogeneity of the study population and contributed to the overall quality of our findings. However, the single-center design of our study may introduce selection bias and limit the generalizability of the findings to other settings.

Moreover, by incorporating subjective, functional, dietary, and anthropometric variables that may influence malnutrition risk, our study offers a comprehensive perspective for future researchers. Finally, although dietary intake data may be subject to recall bias, this potential limitation was minimized through data collection by experienced researchers.

## 5. Conclusions

The prevalence of malnutrition risk was 26.9% in our study group. The risk of malnutrition was higher among individuals with lower appetite, poorer self-perceived health status compared to peers, and smaller waist circumference. Reduced physical function and strength were also associated with an increased risk of malnutrition. These findings emphasize that basic clinical indicators may be important predictors of possible malnutrition risk. These results support the use of simple, non-invasive assessments for early identification of malnutrition risk, facilitating timely interventions. Integrating these measures into routine geriatric care could enhance prevention strategies and improve health outcomes in older adults.

## Figures and Tables

**Figure 1 nutrients-17-02522-f001:**
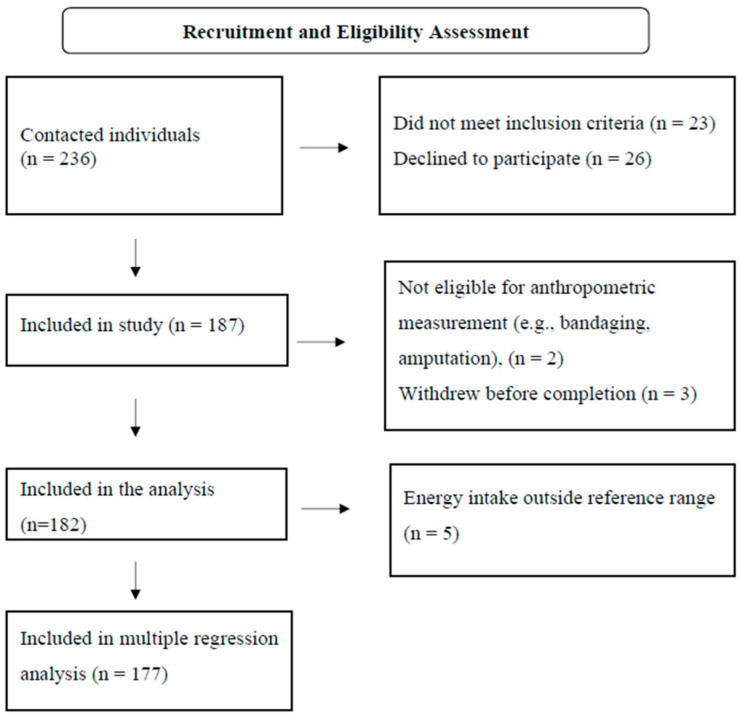
Flowchart for the inclusion of participants in the study.

**Table 1 nutrients-17-02522-t001:** Inclusion and exclusion criteria.

Criteria	Description
Inclusion Criteria	• Being 65 years of age or older
	• Living in the community
	• Having sufficient cooperation and orientation
	• Having the ability to understand and answer questions
Exclusion Criteria	• Bedridden individuals
	• Having a neurological disease and known cognitive dysfunction (such as Alzheimer’s disease, dementia, delirium, etc.)
	• Having a communication disorder and severe hearing problems
	• Having any health problem that can directly cause malnutrition (digestive system diseases, cancer, dysphagia, any pathological condition that can cause chewing-swallowing difficulties)
	• Residing in hospitals or nursing homes
Inclusion Criteria	• Being 65 years of age or older
	• Living in the community
	• Having sufficient cooperation and orientation
	• Having the ability to understand and answer questions
Exclusion Criteria	• Bedridden individuals
	• Having a neurological disease and known cognitive dysfunction (such as Alzheimer’s disease, dementia, delirium, etc.)
	• Having a communication disorder and severe hearing problems
	• Having any health problem that can directly cause malnutrition (digestive system diseases, cancer, dysphagia, any pathological condition that can cause chewing-swallowing difficulties)
	• Residing in hospitals or nursing homes

**Table 2 nutrients-17-02522-t002:** Socio-demographic characteristics of participants by nutritional status.

Variable		Total (*n* = 182)	At Risk of Malnutrition (*n* = 49)	Normal Nutritional Status (*n* = 133)	*p*
x ± SD					
**Age (years)**		72.1 ± 6.0	73.8 ± 6.6	71.5 ± 5.6	0.045
n (%)					
**Age**	65–74	121 (66.5)	26 (53.1)	95 (71.4)	0.038
	75–84	51 (28.0)	18 (36.7)	33 (24.8)	
	85↑	10 (5.5)	5 (10.2)	5 (3.8)	
**Gender**	Female	74(40.7)	29 (59.2)	45 (33.8)	0.004
	Male	108 (59.3)	20 (40.8)	88 (66.2)	
**Education level**	Illiterate ^a^	46 (25.3)	21 (42.9)	25 (18.8)	0.003
	Literate ^b^	29 (15.9)	10 (20.4)	19 (14.3)	
	Primary	66 (36.2)	13 (26.5)	53 (39.8)	
	Secondary	23(12.6)	4 (8.2)	19 (14.3)	
	College	18 (9.9)	1 (2.0)	17 (12.8)	
**Marital status**	Married	148 (81.3)	34 (69.4)	114 (85.7)	0.022
	Separated/Widowed	34 (18.7)	15 (30.6)	19 (14.3)	
**Living Arrangement**	Alone	15 (8.2)	5 (10.2)	10 (7.5)	0.070
	With spouse	67 (36.8)	12 (24.5)	55 (41.4)	
	With nuclear family	61 (33.5)	16 (32.7)	45 (33.8)	
	With extended family	39 (21.5)	16 (32.7)	23 (17.3)	
**Income level (self-reported)**	Income > costs	30 (16.5)	5 (10.2)	25 (18.8)	0.287
	Income = costs	73 (40.1)	19 (38.8)	54 (40.6)	
	Income < costs	79 (43.4)	25 (51.0)	54 (40.6)	
**Social security**	Other	144 (79.1)	32 (65.3)	112 (84.2)	0.010
	Green card ^c^	38 (20.9)	17 (34.7)	21 (15.8)	

^a^: cannot write and read, ^b^: read and write with no formal education, ^c^: health coverage for low-income citizens in Türkiye. SD: standard deviation, n: number.

**Table 3 nutrients-17-02522-t003:** Comparison of health, nutritional, anthropometric, and physical function variables by nutritional status.

Variable		Total (*n* = 182)	At Risk of Malnutrition (*n* = 49)	Normal Nutritional Status (*n* = 133)	*p*
x ± SD					
Perceived health status compared to peers (VAS)		6.1 ± 2.0	5.1 ± 1.8	6.5 ± 1.9	<0.001
Pain intensity (VAS)		4.7 ± 2.9	5.9 ± 2.6	4.3 ± 2.9	0.001
Appetite status (VAS)		6.9 ± 2.2	4.9 ± 2.2	7.6 ± 1.6	0.001
BMI (kg/m^2^)		30.7 ± 5.9	29.1 ± 6.3	31.4 ± 5.6	0.034
Waist circumference (cm)		103.4 ± 12.5	96.6 ± 12.6	105.9 ± 11.6	<0.001
Muscle mass (kg)		48.9 ± 8.6	43.5 ± 6.4	51.0 ± 8.5	<0.001
MUAC (cm)		29.0 ± 3.7	27.8 ± 3.9	29.4 ± 3.6	0.009
Energy intake (kcal) ^a^		1685.0 ± 566.7	1465.0 ± 486.8	1758.8 ± 574.0	0.003
Protein intake (g) ^a^		76.1 ± 34.8	65.8 ± 24.2	79.5 ± 37.1	0.024
n (%)					
**Multimorbidity (2 and above)**	Yes	121 (66.5)	29 (59.2)	92 (69.2)	0.276
	No	61 (33.5)	20 (40.8)	41 (30.8)	
**Polypharmacy (5 and above)**	Yes	47 (25.8)	11 (22.4)	36 (27.1)	0.660
	No	135 (74.2)	38 (77.6)	97 (72.9)	
**MMSE**	Severe	25 (13.7)	14 (28.6)	11 (8.3)	<0.001
	Moderate	88 (48.4)	27 (55.1)	61 (45.9)	
	Mild	18 (9.9)	3 (6.1)	15 (11.3)	
	No cognitive impairment	51 (28.0)	5 (10.2)	46 (34.6)	
**Hand grip strength**	Poor	64 (35.2)	25 (51.0)	39 (29.3)	0.011
	Good	118 (64.8)	24 (49.0)	94 (70.7)	
**Calf circumference**	31 and above	166 (91.2)	39 (79.6)	127 (96.2)	0.001
	Less than 31	15 (8.2)	10 (20.4)	5 (3.8)	
**30-s CST**	10 and above	52 (28.6)	8 (16.3)	44 (33.3)	0.039
	Less than 10	129 (70.9)	41 (83.7)	88 (66.7)	
**TUG Test**	12 s and under	134 (73.6)	32 (65.3)	102 (76.7)	0.175
	Over 12 s	48 (26.4)	17 (34.7)	31 (23.3)	
**Perceived health status compared to peers**	Good	108 (59.3)	17 (34.7)	91 (68.4)	<0.001
	Middle	44 (24.2)	15 (30.6)	29 (21.8)	
	Bad	30 (16.5)	17 (34.7)	13 (9.8)	
**Pain intensity**	Yes	102 (56.0)	38 (77.6)	64 (48.1)	0.001
	No	80 (44.0)	11 (22.4)	69 (51.9)	
**Falls in the last 1 month**	Yes	11 (6.0)	7 (14.3)	4 (3.0)	0.009
	No	171 (94.0)	42 (85.7)	129 (97.0)	
**Appetite status**	Good	127(69.8)	16 (32.7)	111 (83.5)	<0.001
	Middle	32(17.6)	15 (30.6)	17 (12.8)	
	Bad	23 (12.6)	18 (36.7)	5 (3.8)	

^a^: Energy and protein intake analyses were conducted on 177 participants after excluding implausible energy intake values (<800 or >4000 kcal/day for men; <500 or >3500 kcal/day for women). BMI: Body Mass Index, CST: Chair-Stand Test, MMSE: Mini Mental State Examination, MUAC: Mid-Upper Arm Circumference, n: number, SD: standard deviation, TUG: Timed ‘Up & Go’, VAS: Visual Analog Scale.

**Table 4 nutrients-17-02522-t004:** Independent Predictors of Malnutrition Risk: Multivariate Logistic Regression Analysis.

	Univariate Logistic Regression					Multiple Logistic Regression				
	Beta	SE	OR	CI 95%	*p*	Beta	SE	OR	CI 95%	*p*
**Age (years)**	0.06	0.03	1.06	1.01–1.12	0.027					
**Age (groups)**										
65–74	-	-	1	-	-					
75–84	0.69	0.37	1.99	0.97–4.09	0.060					
85↑	1.30	0.67	3.65	0.98–13.59	0.053					
**Gender**										
Female	-	-	1	-	-					
Male	1.04	0.25	2.84	1.45–5.56	0.002					
**Education level**										
Illiterate ^a^	-	-	1	-	-					
Literate ^b^	2.66	1.07	14.28	1.75–116.45	0.013					
Primary	2.19	1.10	8.95	1.04–77.37	0.046					
Secondary	1.43	1.06	4.17	0.51–34.26	0.184					
College	1.28	1.17	3.58	0.36–35.23	0.274					
**Marital status**										
Married	-	-	1	-	-					
Separated/widowed	0.97	0.40	2.65	1.22–5.76	0.014					
**Living Arrangement**										
Alone	-	-	1	-	-					
With spouse	−0.33	0.64	0.72	0.21–2.51	0.604					
With nuclear family	−1.16	0.46	0.31	0.13–0.77	0.011					
With extended family	−0.67	0.44	0.51	0.22–1.20	0.124					
**Social security**										
Other	-	-	1	-	-					
Green Card ^c^	1.04	0.38	2.83	1.34–6.00	0.007					
**Perceived health status compared to peers (VAS)**	−0.37	0.09	0.69	0.58–0.83	<0.001	0.55	0.16	1.73	1.26–2.39	0.001
**Pain intensity (VAS)**	0.19	0.06	1.21	1.08–1.36	0.001					
**Appetite status (VAS)**	−0.63	0.10	0.53	0.44–0.65	<0.001	0.66	0.15	1.94	1.46–2.59	<0.001
**BMI (kg/m^2^)**	−0.07	0.03	0.93	0.88–0.99	0.024					
**Waist circumference (cm)**	−0.06	0.02	0.94	0.91–0.97	<0.001	0.09	0.02	1.09	1.04–1.14	<0.001
**Muscle mass (kg)**	−0.12	0.07	0.88	0.84–0.93	<0.001					
**MUAC (cm)**	−0.13	0.05	0.88	0.80–0.97	0.010					
**Energy intake (kcal) ^d^**	−0.00	0.00	1.00	1.00–1.00	0.001					
**Protein intake (g) ^d^**	−0.02	0.01	0.98	0.97–1.00	0.010					
**MMSE**										
Severe	-	-	1	-	-					
Moderate	2.46	0.62	11.70	3.48–39.45	<0.001					
Mild	1.40	0.53	4.07	1.46–11.39	0.007					
No cognitive impairment	0.61	0.79	1.84	0.39–8.63	0.439					
**Hand grip strength**										
Good	-	-	1	-	-	-	-	1	-	-
Poor	0.92	0.34	2.51	1.28–4.92	0.007	0.07	0.04	1.07	1.00–1.15	0.057
**Calf circumference**										
31 and above	-	-	1	-	-					
less than 31	1.874	0.58	6.51	2.10–20.20	0.001					
**30-s CST**										
10 and above	-	-	1	-	-	-	-	1	-	-
Less than 10	0.941	0.43	2.56	1.12–5.93	0.028	0.33	0.10	1.40	1.15–1.70	0.001
**TUG Test**										
12 s and under	-	-	1	-	-	-	-	1	-	-
Over 12 s	0.56	0.36	1.75	0.86–3.56	0.12	0.24	0.12	1.27	1.01–1.60	0.044
**Perceived health status compared to peers**										
Good	-	-	1	-	-					
Middle	1.02	0.41	2.77	1.23–6.23	0.014					
Bad	1.95	0.45	7.00	2.88–17.02	<0.001					
**Pain intensity**										
Yes	-	-	1	-	-					
No	1.32	0.38	3.72	1.76–7.90	0.001					
**Falls in the last 1 month**										
Yes	-	-	1	-	-					
No	1.68	0.65	5.38	1.50–19.27	0.010					
**Appetite status**										
Good	-	-	1	-	-					
Middle	1.81	0.44	6.12	2.57–14.61	<0.001					
Bad	3.22	0.57	24.98	8.14–76.61	<0.001					

^a^: cannot write and read, ^b^: read and write with no formal education, ^c^: Health coverage for low-income citizens in Türkiye, ^d^: Energy and protein intake analyses were conducted on 177 participants after excluding implausible energy intake values (<800 or >4000 kcal/day for men; <500 or >3500 kcal/day for women). BMI: Body Mass Index, CI: confidence interval, CST: Chair-Stand Test, MMSE: Mini Mental State Examination, MUAC: Mid-Upper Arm Circumference, OR: odds ratio, SE: standard error, TUG: Timed ‘Up & Go’, VAS: Visual Analog Scale.

## Data Availability

The datasets used and analyzed during the current study are available from the corresponding author on reasonable request.

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
