# Peer review of "Prevalence and Determinants of Malnutrition in Community-Dwelling Adults Aged 65 and over in Eastern Türkiye: A Cross-Sectional Study"

_nutrients, 2025, doi:10.3390/nu17152522_

Round 1
Reviewer 1 Report
Comments and Suggestions for Authors
The article targets an important problem, but as it is, it cannot be published. Some of the reasons are:
- being cross sectional, the study does not emphasyse real factor that cause malnutrition
- it is carried out in a single center, so no generalization is possible
- there is a selection bias. Participants were community-dwelling older adults who voluntarily visited a health center. Excluded were those with major illness, cognitive deficits, or living in institutions, exactly the groups in big danger of malnutrition
- the small number of people you worked on had just 3 *three malnourished individuals, you cannot work and draw conclusions on such numbers, even by unifying groups (at risk are not malnourished)
- one day recall for food intake is not at all reliable, especially in elderly
- MMSE was used, but individuals with diagnosed cognitive impairments were excluded. So you ignore the link of cognition with malnutrition
- waist circumference showed a complex relationship with malnutrition, sometimes acting protectively. This reflects the "obesity paradox" in geriatrics, but the study doesn’t adequately explore or control for sarcopenic obesity, which could confound these results.
What you need is a larger, more diverse sample, a longitudinal design, and better dietary assessment tools.
Reviewer 2 Report
Comments and Suggestions for Authors
Dear Authors,
Thank you for your manuscript. Please see my comments below.
Overall, the manuscript "Malnutrition and Its Predictors in Community-Dwelling Older Adults Aged 65 and Over: A Cross-Sectional Study" is well-written and well-structured, providing valuable insights despite the relatively small sample size. The findings significantly contribute to understanding the prevalence and determinants of malnutrition risk among older adults living in community settings. However, there are several technical improvements recommended:
-
Please include the mean age (SD) and gender distribution of the study participants to the abstract to give readers immediate demographic context.
-
The Introduction requires a clearer emphasis on the study's novelty. Explicitly clarify how this research addresses existing gaps in the literature and builds on or diverges from prior studies.
-
Methods Section:
-
Section 2.2 titled "Demographic characteristics" is somewhat misleading, as it also describes health-related variables. Consider retitling this section to better reflect its actual content, such as "Demographic and Health-Related Characteristics."
-
The methods section does not describe all sociodemographic variables analyzed in the Results (e.g., social security status, education level, marital status, living arrangements, and income). These variables should be detailed in the Methods section.
-
Additionally, ethical approval and informed consent processes should be briefly described in the Methods section, not only mentioned at the end in the declarations section.
-
-
Results section. Replace "p = 0.000" with "p < 0.001" throughout the manuscript to adhere to standard statistical reporting guidelines.
-
Table 4 is difficult to read due to excessive decimal places and too large font size. It is recommended to limit the number of decimal places to two and adjust the font size or layout for improved readability.
- The end of the discussion would benefit from a paragraph of practical implications.
Round 2
Reviewer 1 Report
Comments and Suggestions for Authors
Thank you for taking in account our suggestions.